# Deep Brain Stimulation in the Treatment of Tardive Dyskinesia

**DOI:** 10.3390/jcm12051868

**Published:** 2023-02-27

**Authors:** Adrianna Szczakowska, Agata Gabryelska, Oliwia Gawlik-Kotelnicka, Dominik Strzelecki

**Affiliations:** 1Central Teaching Hospital, Medical University of Lodz, 92-213 Lodz, Poland; 2Department of Sleep Medicine and Metabolic Disorders, Medical University of Lodz, 92-215 Lodz, Poland; 3Department of Affective and Psychotic Disorders, Medical University of Lodz, 92-216 Lodz, Poland

**Keywords:** tardive dyskinesia, schizophrenia, antipsychotics, deep brain stimulation

## Abstract

Tardive dyskinesia (TD) is a phenomenon observed following the predominantly long-term use of dopamine receptor blockers (antipsychotics) widely used in psychiatry. TD is a group of involuntary, irregular hyperkinetic movements, mainly in the muscles of the face, eyelid, lips, tongue, and cheeks, and less frequently in the limbs, neck, pelvis, and trunk. In some patients, TD takes on an extremely severe form, massively disrupting functioning and, moreover, causing stigmatization and suffering. Deep brain stimulation (DBS), a method used, among others, in Parkinson’s disease, is also an effective treatment for TD and often becomes a method of last resort, especially in severe, drug-resistant forms. The group of TD patients who have undergone DBS is still very limited. The procedure is relatively new in TD, so the available reliable clinical studies are few and consist mainly of case reports. Unilateral and bilateral stimulation of two sites has proven efficacy in TD treatment. Most authors describe stimulation of the globus pallidus internus (GPi); less frequent descriptions involve the subthalamic nucleus (STN). In the present paper, we provide up-to-date information on the stimulation of both mentioned brain areas. We also compare the efficacy of the two methods by comparing the two available studies that included the largest groups of patients. Although GPi stimulation is more frequently described in literature, our analysis indicates comparable results (reduction of involuntary movements) with STN DBS.

## 1. Introduction

Tardive dyskinesia (TD) is a group of symptoms characterized by irregular and involuntary movements that most commonly affect the tongue, lips, jaw, face, and sometimes the peri-orbital areas. In some cases, patients also have irregular movement of the trunk and limbs [1,2]. Tardive dyskinesia (TD) might be also present as tremor, akathisia, dystonia, chorea, tics, or as a combination of different types of abnormal movements. In addition to movement disorders (including involuntary vocalizations), TD patients may have various sensory symptoms, such as the urge to move (as in akathisia), pain, and paresthesia [3].

TD is a specific type of secondary dystonia, mainly caused by the chronic use of dopamine receptor antagonists. The onset of TD usually occurs after years of taking neuroleptics but may also appear earlier, even after several months. The risk is related, among others, to the strength of the drug binding to the dopaminergic D2 receptor. In the elderly, symptoms may become apparent after a shorter period of use of the drug, the early onset of these symptoms and their intensity may indicate features of organic brain damage [4]. Due to the need for long-term treatment, neuroleptics are the main reason for TD’s appearance in clinical practice. Nevertheless, when using other antidopaminergic drugs such as antiemetics (domperidone, bromopride, and metoclopramide); antidepressants such as trazodone, amitriptyline, clomipramine, fluoxetine; and sertraline or calcium channel blockers, the risk of TD appearance, while significantly lower, should be highlighted [5].

Interestingly, tardive dyskinesia can appear both during the use and after the discontinuation of neuroleptics. The prevalence of tardive dyskinesia is estimated at 0.4–9% in patients receiving antipsychotics, while some studies indicate a more frequent occurrence of TD (20–50%) [6,7]. According to the DSM-5, TD can be diagnosed when antipsychotic-induced tardive dyskinesia follows exposure to neuroleptics for at least three months (one month in individuals aged ≥60 years) and persists for at least one month after the last dose of the drug [8]. This iatrogenic complication may persist long after drug discontinuation and might become permanent [1,6]. TD often results in disability, with mild to severe functional impairment (significantly impaired gait, speech, and swallowing) in about 10% of cases, causing a heavy burden on both patients and their caregivers [6]. In addition to physical burden and pain, tardive dyskinesia leads to social exclusion and ostracism in patients with these symptoms. The involuntary movements typical of TD are a significant burden for patients in a social context, representing one of the archetypal images of mental illness and a reason for stigmatization.

Aside from pharmacological interventions (changing the dose or the drug) or implementing TD-targeted treatment, there is a promising method that may offer new opportunities for this group of patients—deep brain stimulation (DBS). DBS is a clinical procedure in which a precisely controlled electric current is passed through electrodes surgically implanted in the brain. This method enables rapid and, more importantly, long-term improvement in motor function and quality of life (QoL) in patients with TD [1,5].

## 2. Etiology and Risk Factors

It is of key importance that TD has a genetic predisposition, which mediates the risk for TD development [5,9]. Nevertheless, the usage of dopamine receptor antagonists is responsible for the exposure of this predisposition [10,11]. Table 1 shows the factors associated with an increased risk of TD [12,13,14,15,16,17,18,19]. Table 2 summarizes the genetic factors that modulate the risk of TD [20,21,22,23].

The main pathogenetic mechanisms associated with the development of TD are the hypersensitivity of postsynaptic D2 receptors and their upregulation associated with their long-term blockade. This leads to changes in cortico-striatal transmission and motor symptoms [24]. The abnormalities also concern the increase in blood flow in the prefrontal cortex, the anterior cingulate gyrus, and the cerebellum, which accompany the increase in the activity of the prefrontal and premotor cortex during the appearance of involuntary movements, which may indicate a decrease in impulse selection and lead to the appearance of involuntary movements [25]. The constant blocking of D2 receptors along with D1 activation may also be important to explain the appearance of symptoms over a longer period of time and their irreversibility [26]. However, it seems that not only disorders of dopaminergic transmission are involved in the development of TD, but changes in serotonergic, glutamatergic, cholinergic, and opioid transmission may play a supportive role [27,28]. The involvement of the serotonin system in TD is indicated by studies on animal models. It was found that inhibition of serotonergic neurons with 8-OH-DPAT (8-hydroxy-2-(dipropylamino)tetralin significantly reduces TD severity. 8-OH-DPAT is one of the first discovered agonists of the serotonergic 5-HT1A receptors. It mediates hyperpolarization and reduction of the firing rate of the postsynaptic neuron. Conversely, administration of fenfluramine or fluoxetine (both increasing the level of serotonin) suppressed the previously obtained improvement. Preclinical studies indicate that deep brain stimulation of the subthalamic nucleus (STN DBS), a technique described latter in this article, reduced the release of 5-HT in the hippocampus and prefrontal cortex, while deep brain stimulation of the EPN (entopeduncular nucleus, internal globus pallidus (GPi) equivalent in rodents) did not affect 5-HT release. Nevertheless, both STN and EPN DBS attenuate TD with equal effectiveness, despite their different effects on the 5-HT system, leading to the conclusion that the mechanism of 5-HT reduction does not determine the effectiveness of DBS in rats.

Oxidative stress and related neuronal damage both might also participate in the etiology of TD. Antipsychotics, especially classic drugs, may be toxic by directly inhibiting complex I of the mitochondrial electron transport chain. Toxicity may also result from the increased production of free radicals and hydrogen peroxide, which are a consequence of the blockade of the D2 receptor and an increase in dopamine turnover [20,29,30]. The weakening of the antioxidant mechanisms may explain the progressive nature of the changes and their irreversibility [31,32,33]. In neuroimaging studies, a decrease in the caudate nucleus volume was observed in the group of patients diagnosed with schizophrenia with TD compared to those with this psychosis without dyskinesia [10,34,35].

## 3. Assessment Tools

The most widely used instrument to assess TD is the Abnormal Involuntary Movement Scale (AIMS). The patient performs several tasks described in the instructions. On that basis, the severity of facial and oral movements, extremity movements, trunk movements, and global judgments is scored on a 0–4 scale (up to 40 points in total) [36]. A separate evaluation concerns dental status (with an annotation yes/no). Another scale is The Burke–Fahn–Marsden Dystonia Rating Scale (BFMDRS), which consists of movement and disability subscales. This tool measures dystonia in nine body regions (incl. the eyes, mouth/speech and swallowing, neck, trunk, arms, and legs; each extremity is assessed individually) with scores ranging from 0 (lack of symptoms) to 120 [37].

## 4. Pharmacological Treatment

TD treatment is difficult and often leads to disappointing results, so the best method is to prevent its onset [38]. Atypical antipsychotics have a lower potential to cause TD. The drugs should be used in the lowest effective doses, particularly if TD appeared earlier or the current treatment induced its onset. When TD appears, initially, it is necessary to reduce the drug dose or, if this does not eliminate TD, switch to a drug with a lower potential for inducing TD, such as clozapine or quetiapine.

The pharmacological treatment of TD is challenging; conventionally administered pharmacotherapies are only beneficial at the initial stage, and the available data point to a lack of satisfactory outcomes in long-term use [6].

VMAT2 (vesicular monoamine transporter 2) inhibitors: tetrabenazine, valbenazine, and deutetrabenazine are the first drug group recommended for TD treatment [2]. In randomized controlled trials, valbenazine and deutetrabenazine demonstrated efficacy in ameliorating TD symptoms with a favorable benefit–risk ratio. For this reason, valbenazine and deutetrabenazine should be considered a first-line treatments for TD. While the currently available evidence suggests that tetrabenazine is another good option for TD, it is not considered a first-line drug due to greater side effects than other VMAT2 inhibitors and very few studies. Amantadine (300 mg per day) may be used when these treatments are ineffective or contraindicated. However, evidence to support the use of amantadine for TD is scarce and limited to short observations [2]. Another discussed treatment option is the short-term administration of clonazepam, but the effectiveness of this method is also limited. Furthermore, considering the acute and long-term consequences (sedation, cognitive decline, tolerance, addiction, and risk of falls, especially in the elderly), routine use of benzodiazepines is not recommended [2,6]. The use of Vitamin E does not improve TD symptoms but may prevent their worsening. When other options fail, some authors recommend pyridoxine (vitamin B6) use, but the optimal dose and treatment duration has not been established yet [2]. In focal dystonia, such as cervical dystonia, botulinum toxin injection may be applied. It is a highly effective approach, but the level of satisfaction with this treatment is low in some of the patients, and they fail to follow up for repeated injections. Therefore, the pharmacotherapeutic method should be regarded as adjuvant therapy instead of a priority choice (the dose reduction of the TD-inducing drug or change to another drug if possible) as the symptoms progress to the advanced stage [6]. The level B recommendations of the American Academy of Neurology for TD treatment indicate clonazepam, Gingko biloba extract (EGb-761), and diltiazem, while amantadine, tetrabenazine, galantamine, and eicosapentaenoic acid are level C. Other test substances, including reserpine, bromocriptine, biperiden, selegiline, vitamin E, vitamin B6, baclofen, and levetiracetam, have not received a recommendation from the academy at this stage [39]. Newer recommendations position new-generation VMAT2 inhibitors (deutetrabenazine and valbenazine) at level A of recommendation, clonazepam and Ginkgo biloba at level B, while amantadine, tetrabenazine, and GPi DBS (globus pallidus internus deep brain stimulation) are at level C [40]. The American Psychiatric Association (APA) indicates a reversible inhibitor of the VMAT2 (deutetrabenazine and valbenazine as more studied than tetrabenazine) as the first-line treatment for TD [41].

## 5. Deep Brain Stimulation

In recent decades, DBS has been successfully used to treat several movement disorders, including Parkinson’s disease and dystonia. More recently, DBS has also been used to treat patients with tardive dyskinesia and OCD, especially in drug-resistant forms [6,7]. Monopolar (unilateral) stimulation modes are the most commonly used, although we also have descriptions of bipolar mode [42,43,44,45,46]. In addition to the potential for rapid and long-term improvement, the advantages of DBS include its relatively nondestructive nature, adjustability, reversibility, and the ability to perform DBS bilaterally in a single surgical session [6,47].

According to the available studies, this method is safe and minimally invasive, with no severe complications during the follow-up periods [6]. The disadvantages of the DBS technique are the requirement for continuous follow-up visits with repeated optimization of pacing parameters (it can also offer potential parameter adjustments) and the risk of hardware complications (incl. electrode displacement, battery depletion, inflammation around parts of the device) [47]. When the effectiveness of pharmacotherapeutic methods is unsatisfactory and symptoms are chronic and very severe, DBS becomes the treatment of last resort [48].

The primary criterion for inclusion in DBS is a high severity of symptoms that significantly impede function and have lasted for more than a year, with no satisfactory response to pharmacological treatment with clozapine or tetrabenazine for at least four weeks at the highest doses tolerated by the patient. Exclusion criteria are similar to those for patients with other dystonias—significant cognitive impairment, unstable mental status, severe depressive symptoms, and comorbid medical problems that may increase surgical risk; an initial brain scan before the decision on DBS applicability is recommended [45].

In addition to correct patient selection and electrode placement (more effective by image guidance or microelectrode recording implemented in leading centers), proper and time-coordinated programming of the equipment is crucial. This is important because we already have multisegment electrodes (from Abbott/St. Jude, Boston Scientific, and Medtronic), and each segment’s current characteristics can be programmed separately. It complicates programming (current of different amplitude, frequency, amperage, and pulse width can be used) but certainly expands the possibilities for stimulation. Once the electrode has been placed, the adjustment of the electrical field optimizes the clinical outcome. It allows continuous monitoring of the effectiveness of the stimulation and provides an opportunity to implement modifications, but it becomes vital when the initially planned electrode placement has failed (in about 40%). The typical inaccuracy of surgical robots or stereotaxic methods is 1–2 mm. In addition, during surgery, the brain can change position by 2–4 mm, which can be minimized by a staged operation [49,50,51,52,53,54,55,56,57,58]. A similar problem arises when the electrode is displaced. Reprogramming often avoids reoperation and allows optimization of parameters if the dislocation is not critical [59,60]. It is worth adding that no clear guidelines have been developed so far, although there are recommendations regarding the programming of stimulators [61,62,63]. In programming, it is important to be aware of the temporal sequence of observed changes—not all symptoms respond to stimulation simultaneously. For example, during stimulation of the subthalamic nucleus in Parkinson’s disease, first (in seconds) the tremor subsides, followed by rigidity (seconds–minutes), bradykinesia (minutes–hours), and axial symptoms (hours–days). These symptoms appear after the stimulation is turned off in the same order [64,65].

Previous research in TD patients has focused on the stimulation of two areas in the brain: the inner globus pallidus (GPi) and the subthalamic nucleus (STN) belonging to the basal ganglia. These nuclei belong to motor circuits, including cortico-thalamic-basal ganglia junctions, which are believed to be the morphological substrate of TD. Most projects focused on the stimulation of the GPi, the preferred target, while less is known about STN stimulation [4,6]. Nevertheless, both STN and GPi stimulation were shown to be beneficial in reducing TD [38].

### 5.1. Internal Globus Pallidus (GPi)

The primary target of GPi DBS is the posteroventrolateral part [46,47,66,67,68,69]. Several descriptions concern the stimulation of the posteroventromedial area [70,71]. Ventral parts of the posterior globus pallidus have a somatotopic organization associated with the motor cortex, which determines the goals of stimulation; the median part is related to the limbic cortex, while the dorsal area is associated with the prefrontal cortex [72].

Stereotactic techniques based on MRI (magnetic resonance imaging) or CT-MRI (a combination of CT and MRI techniques) help correct electrode placement [73]. The optimal electrode placement is typically within 19–22 mm lateral to the line between the anterior and posterior commissure, 4–6 mm inferior to that line, and 2–4 mm anterior to the mid-commissural point [45,46,67,71,74,75,76,77,78,79]. In one description, the electrode position corresponded to the somatotopic face area [80]. The most common practice uses microelectrode recordings (MERs) to detect discharges of neurons in the GPi and to order “noisy signals” with DBS. The most common stimulation parameters used were the voltage (amplitude) of the current (1.0–7.0 V) [43,67], frequency (60–185 Hz) [42,69,78,81], and pulse width (60–450 µs) [42,45,78,81,82,83]. A detailed list of electrodes used, voltages, location, and effectiveness of the treatments can be found in the study by Morigaki et al. [84]. With several exceptions of bipolar modes [42,43,44,45,46], other reports concern monopolar stimulations.

Much of the literature was single-patient reports [43,47,68,70,73,74,75,77,78,80,82,85,86,87,88,89], small groups of 2–4 people [46,48,67,69,71,79,90,91], or slightly larger groups [42,45,76,81,83,92,93,94,95], and 19 patients comprised the largest cohort [38] (Table 3).

#### 5.1.1. Motor Effects of GPi DBS

The reported efficacy (reduction in dystonia scores) ranges from 28% to 100%, with most reports showing ≧60% improvement, with a follow-up period of up to 11 years [38]. Improvement is described as stable even after 4-year follow up. In addition to improvement in symptoms, most investigators consistently report a significantly favorable change in the quality of life and daily functioning. Nevertheless, there are also descriptions of no overall change in this area [45,96].

Clinical responses appear either during the surgical procedure and the first activation of stimulation or in the first days after turning on the equipment [45,46,67,68,70,76,86,91]. If clinical responses are observed shortly after switching on the device, we can precisely program the equipment at the outset; in other cases, patient adjustments are carried out at follow-up visits or via the Internet, more recently [97]. The manufacturer recommends the lowest sufficient stimulator settings, combining optimal performance with less load and then longer battery life or less frequent recharging.

Changes in the treatment of choreiform dyskinesia are noted earlier, tonic postural dystonia responds later, symptoms improve gradually, and changes are observed after weeks or even months of stimulation [44,46,75,86,91,92,93]. In fixed dystonias, the efficacy of GPi DBS is lower [42,45,67,81].

#### 5.1.2. Side Effects of GPi DBS

Despite its invasiveness, DBS is characterized by a low number of complications and is considered a safe, effective, and well-tolerated method [4]. The frequency of all side effects reaches 9%. Observations of nonmotor effects are very rare. DBS may induce transient affective states (mild to moderate depressive syndrome in most cases); the authors also emphasized some increase in suicidal risk [73,98]. However, at longer follow up, there was an improvement in mood, which could also be explained by relief from the burden of motor symptoms, disability, or social impact [38,45,76,80,99]. In one study, six months after treatment, one patient had a brief psychotic episode, and another patient had symptomatic improvement allowing the discontinuation of antipsychotic drugs [76]. Contrary to the first reports, the negative influence of continuous pallidal (GPi) DBS on cognitive functions has not been confirmed [38,45,71], while one study notes improvement [99].

The procedure of implanting the electrode (in both locations, GPi and STN) is associated with the possibility of incorrect placement or electrode displacement, infections, pain associated with the connection cable, intracranial hemorrhage, and seizure. Gait and balance disturbances contributing to falls have also been observed. These disturbances were transient and resolved after the optimization of DBS parameters [38]. The GPi is involved in speech fluency; thus, slowing, halting, and imprecise oral articulation and reduced voicing control are common symptoms during DBS in this area. Bilateral DBS induces more speech difficulties [100]. Dysarthria occurs in almost 30% of patients; severe cases may require speech therapy [38]. Despite the complications being infrequent, the risk–benefit ratio always needs to be weighed. DBS becomes the last resort in patients with severe TD when symptoms are severe, functioning is significantly impaired, and other treatment options are insufficient. Table 4 shows the most common side effects, along with the structures whose stimulation is responsible for their appearance.

### 5.2. Subthalamic Nucleus (STN)

The subthalamic nucleus (STN), belonging to the basal ganglia, was the first neurosurgical target in the treatment of dystonia (thalamotomy), but data about STN DBS in treating TD are still scarce. Less frequent use is, among others, related to psychiatric complications (depression, suicidality, mania, and impulse-control problems) observed during DBS of this brain structure in patients with Parkinson’s disease. The best control of motor symptoms is provided by stimulation of the sensorimotor (dorsolateral) area of the STN [101].

#### 5.2.1. Motor Symptoms of STN DBS

So far, only a limited number of cases of STN DBS for TD have been reported. In addition to the Deng study, which we will discuss later [6], Zhang et al. published a description of a series of nine patients treated with STN DBS for secondary dystonia (two with tardive dystonia) [102]. In one case, the dystonia following neuroleptic treatment improved by 92% in the BFMDRS 3 months after stimulator implementation. Long-term observation of one of those patients with severe TD dystonic symptoms initially is described by Meng et al.; the patient had no neurological symptoms after 144 months (6 and 12 years after the operation BFMDRS total score was 0) [4]. Another study (12 patients with primary dystonia and 2 with TD) using STN DBS showed improvement ranging from 76 to 100% in the BFMDRS [103]. One patient underwent DBS electrode placement in the left and right STN with a near-complete resolution of tremors [104] (data summarized in Table 5). 

#### 5.2.2. Side Effects of STN DBS

The anatomical location of the STN is very close to several functionally significant areas. Therefore, the induced side effects are also associated with stimulating adjacent nuclei and nerve tracts. Table 6 presents the most common side effects with the postulated structures responsible for their appearance. Due to the lack of detailed descriptions regarding TD, the table lists observations during STN DBS in Parkinson’s disease.

### 5.3. Internal Globus Pallidus (GPi) and Subthalamic Nucleus (STN) DBS Comparison

Authors suggest better results for STN DBS using lower stimulation parameters than in GPi DBS, but no studies compared the effects of DBS in the two areas. In the following section, we will compare the results of two studies of the GPi and STN involving the largest groups of TD patients.

The largest study evaluating the efficacy of GPi DBS is by Pouclet-Courtemanche et al. It originally included 19 patients, while 18 reached a 6-month follow up, 14 participants were assessed at long-term follow up (6–11 years) [38]. Meanwhile, Deng et al. analyzed STN DBS results in a group of 10 patients, with all included evaluations at 6 months and long-term follow up (12–105 months) [6]. The aforementioned time points were common for both studies among other follow-up lengths. Furthermore, the mutual form of assessment of motor symptoms was only the AIMS. We compared the effectiveness of DBS at the different sites using a two-sample z-test for proportions. In the case of the study by Pouclet-Courtemanche et al., no median/mean data for the AIMS score were available at all time points. Regardless, the calculation of proportions was possible based on the graph analysis presenting a change in the AIMS score at the different follow ups. For the 6-month follow-up time point, the proposition was 0.49 (*n* = 18) and 0.15 (*n* = 10) for the GPi DBS and the STN DBS, respectively. In the comparison, the difference did not reach statistical significance with *p* = 0.079, mostly due to the small sample sizes in both studies, as the trend is visible (Figure 1). We did perform a statistical analysis of a long-term follow up due to a disparity in the observation period, which could affect the result.

## 6. Discussion

Deep brain stimulation (DBS) is an established treatment for patients with tardive dyskinesia when pharmacological therapy alone does not provide sufficient relief or is associated with disabling side effects. With this method, patients achieve satisfactory results in both the short and long term, with a relatively small number of complications. As we previously mentioned, the main sites with proven efficacy of stimulation are the subthalamic nucleus (STN) and internal globus pallidus (GPi). Although the GPi remains the standard stimulation target, our comparison in small groups shows at least comparable efficacy of STN and GPi DBS, including 6-month follow up. Similar conclusions come from comparisons of the two methods in PD [105]. However, further research is needed to confirm this conclusion, also because the trend may indicate an advantage for STN DBS. DBS studies in PD allow some conclusions that may also apply to the treatment of TD with this method. The advantage of GPi stimulation lies in the possibility of effective use of the electrode unilaterally and somewhat easier optimization of current parameter programming. On the other hand, some researchers report that STN DBS may be less likely to cause adverse symptoms in mood, cognitive function, gait, and speech [106]. The GPi is occasionally indicated as the preferred target in treating oral TD and dystonia, while STN DBS could be considered an effective and safe procedure in patients with predominant tardive Parkinsonism and/or tardive tremor [104].

In contrast, studies by Sun et al. indicate some advantages of STN DBS stimulation in dystonias, including TD. According to these authors, symptomatic improvement begins immediately after stimulation, which allows for a quick selection of the best stimulation parameters. The stimulation parameters used for the GPi are higher than those used during STN DBS, resulting in longer battery life for STN DBS (longer intervals between charges). According to the authors, STN DBS results in better symptomatic control than GPi DBS in dystonia patients (compared to data obtained by other teams) [103].

To broaden knowledge and outline plans for necessary research, it is worth looking at solutions employed in DBS procedures in patients with other health problems. DBS is a method that has been implemented for years in various conditions such as dystonia, Parkinson’s disease, and obsessive–compulsive disorder. This method is also recommended for patients with severe and treatment-resistant forms of the disease. It is noteworthy that the STN is the standard site of stimulation in PD [107]. According to the symptomatic profile of PD, preferences include alternative targets, e.g., the thalamic ventral intermediate nucleus (VIM) or the GPi. Recent research in this area has focused on the search for other sites of stimulation such as the posterior subthalamic area (PSA) or the caudal zona incerta (cZi). The PSA is located ventrally to the VIM, between the red nucleus and the STN. PSA DBS is not significantly different from VIM DBS in suppressing tremor, but clinical benefit from PSA DBS is attained at lower stimulation amplitudes [108]. Furthermore, several open-label studies have shown a good effect in the reduction of PD symptoms with DBS in the caudal zona incerta (cZi) [109].

While both TD and PD treatment have the same standard stimulation sites, it is worth investigating other experimental stimulation sites in TD treatment, such as the PSA or the cZi, or finding new targets. Treatment of refractory TD with DBS is not a low-cost method, requiring an experienced neurosurgical team and precise instrumentation. It is also not a life-saving method, but, if we want to have a full range of possible medical procedures that may expand our understanding of the brain (we consider it crucial), this research must be continued and intensified. The latest technical achievements in the field of construction of stimulators and electrodes, e.g., modeling the shape of the impact field, as well as the results of new studies focused on the paths connecting the gray matter of various brain regions allow us to expect discoveries in research using DBS, hopefully also in TD.

## Figures and Tables

**Figure 1 jcm-12-01868-f001:**
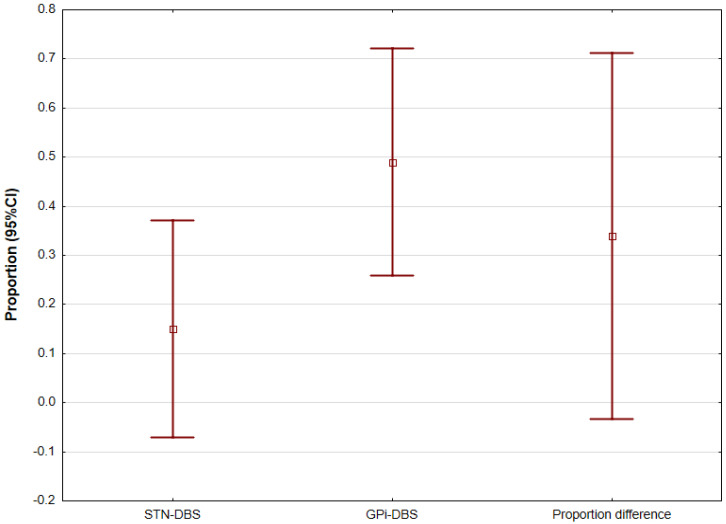
Proportion comparison of AIMS score (initial evaluation to 6-months follow up) between STN DBS and GPi DBS. AIMS—Abnormal Involuntary Movement Scale, DBS—deep brain stimulation, GPi—internal globus pallidus, STN—subthalamic nucleus.

**Table 1 jcm-12-01868-t001:** Nonmodifiable and modifiable risk factors of TD.

Nonmodifiable Factors	Modifiable Factors
Advanced age	Type of dopamine receptor blocking agents
Female sex	Duration of illness
Caucasian or African ethnicity	Dosage and length of exposure to a dopamine receptor blocker
Intellectual disability	Intermittent antipsychotic treatment
Brain damage	Anticholinergic treatment
Negative symptoms in schizophrenia	Smoking
	Alcohol and cocaine abuse/dependence
	Akathisia

**Table 2 jcm-12-01868-t002:** Genes whose polymorphisms increase the risk of TD.

DRD2 and DRD3
HTR2A (5-HT2A receptors)
COMT
MnSOD
Cytochrome P450 (CYP2D6)
GSK-3ß
3′-Regulatory region of Nurr77 mRNA
SLC6A11, GABRB2, and GABRG3 related to GABAergic transmission
GRIN2A related to NMDA receptor and glutamatergic transmission
GSTM1, GSTP1, NOS3, and NQO1 involved in oxidative stress reactions
BDNF
GLI2
HSPG2

Genes DRD2 and DRD3—D2 and D3 receptor, D-dopamine; HTR2A-5—hydroxytryptamine receptor 2A, 5-HT–serotonin; COMT—catechol-O-methyl-transferase; MnSOD—manganese super dismutase; CYP2D6—cytochrome P450 2D6; GSK2ß—glycogen synthase kinase 2 beta; mRNA—messenger RNA; SLC6A11—solute carrier family 6 member 11; GABRB2—gamma-aminobutyric acid type A receptor subunit beta 2; GABRG3—gamma-aminobutyric acid type A-rho receptor subunit gamma 3; GABA—γ-aminobutyric acid; GRIN2A—glutamate ionotropic receptor NMDA type subunit 2A; NMDA—N-methyl-D-aspartate; GSTM1—glutathione S-transferase Mu 1; GSTP1—glutathione S-transferases P1; NOS3—nitric oxide synthase 3; NQO1—NAD(P)H quinone dehydrogenase 1; BDNF—brain-derived neurotrophic factor; GLI2—GLI family zinc finger 2; HSPG2—heparan sulfate proteoglycan 2.

**Table 3 jcm-12-01868-t003:** Basic parameters and outcomes from GPi DBS studies.

Author [Reference]	Localization	Mono-/Bipolar (N, When >1)	Scale (% of Improvement)/Follow Up (Months)
Pouclet-Courtemanche [38]	PV-GPi	M	AIMS (63)/12–132
Sako [42]	PV-GPi	M/B (5)	BFMDRS-M (58–100), BFMDRS-D (67–100)/3–49
Nandi [43]	PV-GPi	B	BFMDRS-M (28), BFMDRS-D (39), AIMS (42)/ 12
Gruber [45]	PVL-GPi	M/B (8)	BFMDRS-M (64-100), BFMDRS-D (25–100), AIMS (33–100)/26–80
Capelle [46]	PVL-GPi	B (4)	BFMDRS-M (70–91), BFMDRS-D (50–100)/16–36
Kim [47]	PVL-GPi	M	BFMDRS-M (97), BFMDRS-D (100)/20
Sobstyl [48]	PVL-GPi	B (2)	BFMDRS-M (69–78), BFMDRS-D (56–73)/12–24
Franzini [67]	PVL-GPi	M (2)	BFMDRS-M (86–88)/12
Kovacs [68]	PVL-GPi	?	BFMDRS-M (97), BFMDRS-D (96)/12
Starr [69]	PVL-GPi	? (4)	BFMDRS-M (6–100)/9–27
Trottenberg [70]	PV-GPi	M	BFMDRS-M (73), AIMS (54)/6
Hälbig [71]	PVM-GPi	M (2)	BFMDRS-M (77–93)/?
Spindler [73]	GPi	M	AIMS (67)/<60
Magariños-Ascone [74]	GPi	?	BFMDRS-M (48), BFMDRS-D (44)/12
Eltahawy [75]	PV-GPi	M	BFMDRS-M (60)/18
Trottenberg [76]	PVM-GPi	M (5)	BFMDRS-M (75–98), BFMDRS-D (80–100)/6
Katsakiori [77]	GPi	M	BFMDRS-M (94), BFMDRS-D (84)/12
Kefalopoulou [78]	GPi	M	BFMDRS-M (91), AIMS (77)/6
Krause [79]	GPi	M (3)	BFMDRS-M (−1–0), no benefit/≤36
Kosel [80]	GPi	M	BFMDRS-M (35)/18
Shaikh [81]	GPi	M (8)	BFMDRS-M (67–100)/6–60
Schrader [82]	GPi	M	AIMS (63)/ 5
Egidi [83]	GPi	M	BFMDRS-M (47), BFMDRS-D (55)/?
Pretto [85]	GPi	B	BFMDRS (~90)/6
Boulogne [86]	PVL-GPi	M	AIMS (79)/120
Trinh [87]	GPi	?	BFMDRS-M (90), BFMDRS-D (87)/18
Puri [88]	GPi	?	AIMS (55)/6
Ogata [89]	PL-GPi	B	BFMDRS-M (69), BFMDRS-D (64), AIMS (94)/7
Woo [90]	PV-GPi	M (3)	BFMDRS-M (54–100)/3–120
Cohen [91]	GPi	M (2)	BFMDRS-M (63–88), BFMDRS-D (53–100)/7–13
Damier [92]	PVL-GPi	M (10)	AIMS (33–78)/6
Chang [93]	PV-GPi	M	BFMDRS-M (71), BFMDRS-D (48), AIMS (77)/27–76
Krause [94]	GPi	B (7)	BFMDRS-M (90), BFMDRS-D (79), AIMS (73)/63–171
Koyama [95]	GPi	B (12)	BFMDRS (78)/6–186

GPi—internal globus pallidus; DBS—deep brain stimulation; PV—posteroventral, PVL—posteroventral lateral; PVM—posteroventral medial; PL—posterolateral; AIMS—Abnormal Involuntary Movement Scale; BFMDRS-M—Burke–Fahn–Marsden Dystonia Rating Scale, movement subscale; BFMDRS-D—Burke–Fahn–Marsden Dystonia Rating Scale, disability subscale; BFMDRS—Burke–Fahn–Marsden Dystonia Rating Scale, total score; ?—data not provided.

**Table 4 jcm-12-01868-t004:** Side effects of GPi DBS and the areas whose stimulation is responsible for these symptoms.

Side Effect	Brain Area
Mood and cognitive symptoms	Ventral part of GPi
Motor side effects (corticospinal and corticobulbar side, i.e., tonic muscle contractions)	Posterior part of GPi/capsular fibers
Phosphenes (seeing light without light entering the eye)	Ventral/optic tract
Low threshold for capsular side effects (i.e., muscle contractions)	Medial GPi
Speech impairment	Internal capsule, medial and posterior to GPi

GPi—globus pallidus internus.

**Table 5 jcm-12-01868-t005:** Basic parameters and outcomes from STN DBS studies.

Author [Reference]	Localization	Mono-/Bipolar (N, When >1)	Scale (% of Improvement) /Follow Up (Months)
Deng [6]	STN	B (10)	BFMDRS (88), AIMS (94)/12–105
Zhang [102]	STN	B (2)	BFMDRS (>90)/3–36
Sun [103]	STN	B (2)	AIMS (63) BFMDRS (>77)/6–42
Kashyap [104]	STN	B	?, “near-complete resolution of tremors”/24

STN—subthalamic nucleus; DBS—deep brain stimulation; BFMDRS—Burke–Fahn–Marsden Dystonia Rating Scale, total score; AIMS—Abnormal Involuntary Movement Scale; ?—data not provided.

**Table 6 jcm-12-01868-t006:** Side effects and the brain area surrounding STN, which stimulation may be responsible for the appearance of symptoms.

Side Effect	Brain Area
Spastic muscle contraction	Internal capsule
Uni- or bilateral gaze deviation	Fibers stemming from the frontal eye field running in the internal capsule, fibers of the third nerve (inferomedial to the STN and within the red nucleus), sympathetic fibers within the zona incerta or STN
Autonomic symptoms	Hypothalamus and red nucleus
Paresthesia	Medial lemniscus
Speech impairment	Internal capsule, the pallidal and cerebello-thalamic fiber tracts medial and dorsal of the STN, medial left-sided STN stimulation in right-handed patients, higher left STN voltage
Depression	Substantia nigra
Mania	Medial and ventral areas of STN
Impulse control disorder	Ventromedial and limbic areas of STN, SNr, medial forebrain bundle
Cognitive problems	Ventral and medial parts of STN, perforation of the caudate nucleus during surgery

STN—subthalamic nucleus, SNr—substantia nigra pars reticulata.

## Data Availability

Not applicable.

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
