# Peer review of "Deep Brain Stimulation in the Treatment of Tardive Dyskinesia"

_jcm, 2023, doi:10.3390/jcm12051868_

Round 1
Reviewer 1 Report
In this manuscript, the authors present a compilation of information related to the use of deep brain stimulation (DBS) in the treatment of tardive dyskinesia (TD). The goal of the manuscript appears to be to "provide up-to-date information about stimulation" of STN and GPi and to compare the largest studies in each target. The authors state that GPi stimulation is more frequently used, but the basis for this statement is not clear.
In the abstract, please change "hypothalamic nucleus" to "subthalamic nucleus". this term appears in other areas of the manuscript and should be changed (eg lines 209, 214).
Consider tempering the statements in lines 53-55, perhaps limiting the statement to "Involuntary movements in TD often draw attention to them and the possibility of underlying psychiatric disease." (or similar)
Table 2 lists genes associated with risk of TD. The table would benefit from further explanation of the genetic abnormality (ie, point mutations, polymorphisms, copy number, etc) and how it modifies risk of TD (ie increase, decrease).
Missing from the introductory sections is a clear description of the various phenomenologies of tardive dyskinesia, which is important when considering candidacy for DBS as well as outcomes.
Many of the sections are quite long and/or contain a level of detail and extra facts that detract from the main focus of the article. These sections could also be better organized so that the information provided flows more naturally and builds on previous statements.
The authors may wish to cite the American Psychiatric Association 2021 recommendations about first-line use of VMAT2 inhibitors for treatment of TD in schizophrenia.
They should also note that all manufacturers, including Medtronic, have segmented/directional leads (line 194).
This review article would be strengthened by a table summarizing key details about the various reports cited in lines 238-240.
A table similar to Table 3 but focused on side effects of GPi DBS should also be provided.
The terminology "non-motor effects" is a bit of a misnomer when it relates to DBS for TD, as TD is not really considered to have non-motor symptomatology. What the authors mainly discuss in this arena are side effects and so section headers could be restricted to that.
Section 5.3 should be omitted. The selection criteria for comparison of the two studies is not defined. The statistical methods are also not ideal. More relevant details about each clinical study could be incorporated into the appropriate preceding sections.
Ultimately, the reader is not left with a good understanding of why DBS in either target is effective in treating TD, why or when it should be used, or what results to expect. The conclusions drawn and suggestions made in the discussion section are not well justified.
Reviewer 2 Report
Title: Deep brain stimulation in the treatment of tardive dyskinesia
Summary:
The authors provide a nice review of the current treatment approach to tardive dyskinesia with an emphasis on the present knowledge of deep brain stimulation efficacy in symptomatic control.
Critique:
Abstract
- Minor grammatical issues
1. Introduction
- It is odd that DBS is not mentioned in your introduction since this is a review on its application in TD
2. Etiology and risk factors
- Is akathisia a modifiable risk factor for TD?
- The last paragraph seems out of place.
5. Deep brain stimulation
- The statement “an initial brain scan is recommended.” At this stage in pre-operative DBS image-guided technology, a high resolution MRI of the brain is necessary for high precision lead placement.
- “This complicates programming 195 (current of different amplitude, voltage, amperage, and pulse width can be used) but certainly expands the possibilities for stimulation.” Frequency should be listed and voltage can be removed since current systems are constant current.
- “Once the electrode has been placed, the direction of the electrode can be changed, optimizing the clinical outcome.” This is incorrect. I think you mean to say that the direction of the current. “adjustment of the electrical field” is a more appropriate.
- How is this continuous monitoring?
- Electrode placement failure varies from institution to institution likely due to surgical technique and protocols (ie. Image-guidance, microelectrode recording, etc). These should be mentioned
- Brain shift is a reason for staged DBS.
- What do you mean by electrode displacement?
5.1 GPi
- Listed electrode placement coordinates are a little restrictive. Suggest saying “…optimal electrode placement is typically within…”
5.1.2 Non-motor effects & side effects
- Avoid use of “stuttering.” Slow, halting, etc. are better speech descriptors. Stuttering gives little localizing value.
- No mention of unilateral vs. bilateral risk of speech disturbance. Presumably greater for bilateral. Have all reports of TD treated with DBS been bilateral?
5.3 GPi vs. STN
- Battery life is of little concern now with rechargeables so this doesn’t seem to be a compelling reason to choose one target over another especially when other efficacy/side effect data is so limited.

Round 2
Reviewer 1 Report
This revised version of the manuscript is improved. While most of the initial comments are adequately addressed, I have the following suggestions for further improvements:
The authors should consider rebalancing how much general information about DBS is being presented in this manuscript, compared to the amount focused particularly on Gpi or STN DBS for TD. For example, much of the information in section 5 on DBS is informative but is not specific to TD. if the authors can trim this down to focus on what is currently understood about these issues with relation to DBS for TD, it will add more value to the reader. Similarly, table 7 and the broadened comments about efficacy of STN vs GPi in patients with Parkinson's disease are not applicable to this topic and should be removed (lines 477-498).
Table 3 - Please change the title of the table to "GPi DBS" (not STN). Also it should include a column with the Ns of patients and duration of follow-up. The relevance of including the column with monopolar vs bipolar stimulation is uncertain and could be removed in favor of the other 2 suggested columns.
similar comments about table 5
Author Response
Dear Reviewer, we want to thank you for your comments, we really appreciate it.
The authors should consider rebalancing how much general information about DBS is being presented in this manuscript, compared to the amount focused particularly on GPi or STN DBS for TD. For example, much of the information in section 5 on DBS is informative but is not specific to TD. if the authors can trim this down to focus on what is currently understood about these issues with relation to DBS for TD, it will add more value to the reader.
As psychiatrists, we regard the article as an introduction to a very interesting method that is not used very often, and as such, it needs a rather detailed introduction, starting with the basic data. We have discussed your comments in the team, and if possible, we would like to keep this part unchanged, however. Thank you for your understanding.
Similarly, table 7 and the broadened comments about efficacy of STN vs GPi in patients with Parkinson's disease are not applicable to this topic and should be removed (lines 477-498).
We have removed table 7 and shortened this part of the article. We have also added more TD-specific information in the discussion. Thank you.
Table 3 - Please change the title of the table to "GPi DBS" (not STN). Also it should include a column with the Ns of patients and duration of follow-up. The relevance of including the column with monopolar vs bipolar stimulation is uncertain and could be removed in favor of the other 2 suggested columns. Similar comments about table 5.
We have left the layout of the two tables adding data on the follow up period. Title of table 3 is correct now. Thank you.
We have also made several language changes, and we hope they will find your approval.
With best regards,
Dr. Adrianna Szczakowska
Dr. Agata Gabryelska
Dr. Oliwia Gawlik-Kotelnicka
Prof. Dominik Strzelecki